# Diagnosed prevalence of Ehlers-Danlos syndrome and hypermobility spectrum disorder in Wales, UK: a national electronic cohort study and case–control comparison

Joanne C Demmler ,[1] Mark D Atkinson,[1] Emma J Reinhold,[2] Ernest Choy,[3] Ronan A Lyons ,[1] Sinead T Brophy[1]

[1]Swansea University Medical School, Swansea University, Swansea, UK
[2]Royal College of General Practitioners, London, UK
[3]School of Medicine, Cardiff University, Cardiff, UK

**Correspondence to**
Dr Joanne C Demmler;
j.demmler@swansea.ac.uk

Dr Emma J Reinhold;
e.reinhold@doctors.org.uk

## ABSTRACT

**Objectives** To describe the epidemiology of diagnosed hypermobility spectrum disorder (HSD) and Ehlers-Danlos syndromes (EDS) using linked electronic medical records. To examine whether these conditions remain rare and primarily affect the musculoskeletal system.

**Design** Nationwide linked electronic cohort and nested case–control study.

**Setting** Routinely collected data from primary care and hospital admissions in Wales, UK.

**Participants** People within the primary care or hospital data systems with a coded diagnosis of EDS or joint hypermobility syndrome (JHS) between 1 July 1990 and 30 June 2017.

**Main outcome measures** Combined prevalence of JHS and EDS in Wales. Additional diagnosis and prescription data in those diagnosed with EDS or JHS compared with matched controls.

**Results** We found 6021 individuals (men: 30%, women: 70%) with a diagnostic code of either EDS or JHS. This gives a diagnosed point prevalence of 194.2 per 100 000 in 2016/2017 or roughly 10 cases in a practice of 5000 patients. There was a pronounced gender difference of 8.5 years (95% CI: 7.70 to 9.22) in the mean age at diagnosis. EDS or JHS was not only associated with high odds for other musculoskeletal diagnoses and drug prescriptions but also with significantly higher odds of a diagnosis in other disease categories (eg, mental health, nervous and digestive systems) and higher odds of a prescription in most disease categories (eg, gastrointestinal and cardiovascular drugs) within the 12 months before and after the first recorded diagnosis.

**Conclusions** EDS and JHS (since March 2017 classified as EDS or HSD) have historically been considered rare diseases only affecting the musculoskeletal system and soft tissues. These data demonstrate that both these assertions should be reconsidered.

## INTRODUCTION

The Ehlers-Danlos syndromes (EDS) are a group of hereditary connective tissue disorders which mainly affect collagen. The nomenclature of these conditions has undergone a

## Strengths and limitations of this study

► Large cohort and nested case–control studies based on whole population routinely collected health data from primary and secondary care.
► We are unable to quantify how many people are suffering from hypermobile Ehlers-Danlos syndrome (hEDS) or hypermobility spectrum disorder but remain undiagnosed, nor can we make any statement on the reliability of the diagnoses.
► Although we only compared codes at Read chapter level, all diagnoses and prescriptions can be matched to conditions found in the EDS/joint hypermobility syndrome literature.

number of iterations which makes discussion of their prevalence complicated (box 1).

For many decades, studies have quoted a prevalence rate of 1 in 5000 for EDS, although the origin of this figure is unclear, seeming to appear first in a medical textbook[1 2] as an unreferenced 'reasonable estimate'. Thus, these syndromes have long been categorised as rare diseases, defined in the European Union as those affecting fewer than 50 in 100 000 people.[3] Kulas Søborg *et al*[4] recently reported a prevalence of 20 per 100 000 for EDS in a nationwide Danish cohort based on secondary healthcare data up to 2012, but importantly, this data did not include patients who had received the considerably more common joint hypermobility syndrome (JHS) diagnosis, now included in the latest revised classification. It is possible to extrapolate a combined population prevalence figure for JHS and EDS for Sweden[5] of around 120 per 100 000 from a study focusing on comorbid mental health issues, but no investigators have thus far set out to investigate the

**Box 1   An overview of the Ehlers-Danlos nomenclature**

► Joint hypermobility per se is reasonably common and thought to be present in around 10% of the general UK population.[40]
► The Brighton criteria were used to diagnose joint hypermobility syndrome (JHS) from 1998.[41]
► The Villefranche criteria were applied to confirm EDS-hypermobility type (EDS-HT) from 1997.[42]
► Prior to the Villefranche criteria, the diagnosis EDS III was used to denote the hypermobile subtype of EDS.
► It was recognised over a number of years that JHS and EDS-HT were not distinct from one another.[43]
► In March 2017, the International Consortium on the Ehlers-Danlos syndromes published a revised classification[44] naming two syndromes:
  Hypermobile EDS (hEDS) which has narrowly defined criteria.
  Hypermobility spectrum disorder (HSD) for those with some but not all the features of hEDS.
► Patients who have a diagnosis of EDS-HT or JHS will fall into one of these two new categories.
► Castori *et al* showed that patients may move from the HSD category into hEDS over time: they also emphasised that the approach to management and the prognosis in terms of disability are the same.[45] One may therefore conclude that health needs across these groups are similar.

combined diagnosed prevalence of JHS/EDS within a population.

Although common features of these conditions are arthralgia, soft tissue injury and joint instability,[6] over the last two decades it has become clear that their clinical features are not limited to musculoskeletal and cutaneous involvement, but are multisystemic.[7–9] In the special edition of the *American Journal of Medical Genetics* dedicated to EDS in March 2017, papers covered links to cardiovascular autonomic[10] and gastrointestinal dysfunction[11] as well as psychiatric and neurodevelopmental disorders.[5 12] Chronic disabling fatigue[13] and pain syndromes[14] were also recognised as common and multifactorial issues. Gynaecological[15 16] and obstetric[17] issues are also reported in this population. There is also an emerging link with the potentially life-threatening condition of mast cell activation syndrome.[18 19] There is some emerging evidence hinting that nutritional deficiencies[20 21] may play a key role, both seeming to be more prevalent in these patients and possibly implicated in the development of some of the complications.

Early diagnosis is found to be crucial to patients[22] to enable the provision of appropriate treatment, as well as to prevent later onset complications.[7] Establishing the diagnosis of EDS/hypermobility spectrum disorder (HSD) is often problematic for patients, which interferes with the early detection, treatment and prevention of further escalations of recognised symptoms, disability and more elaborate complications. A mean of 14 years elapses between the first clinical manifestations and the actual diagnosis.[23] For 25% of patients, this delay lasts over 28 years.[23] "A misdiagnosis was given to 56% of patients [resulting in] inappropriate treatment in 70% of the patients…. For

86% of the patients, the delay in diagnosis was considered responsible for deleterious consequences."[23(p.137)]

It is possible that some of these difficulties arise from the widespread belief among clinicians that EDS is rare. It is therefore of clinical importance to establish better estimates of current prevalence. Conventional studies tend to be based in restricted clinical settings, such as rheumatology clinics, and are therefore limited by the number of recruited patients and biased by severity/type of patients referred. It has been shown that using linked health data is an economic and effective alternative to performing de novo longitudinal studies, including rare conditions.[24 25] We used routinely held data from primary and secondary care sources to examine the epidemiology of people with a diagnostic code for EDS/JHS in Wales. We then conducted a nested case–control study to study the number of diagnoses across all body/disease systems and prescription usage to test the widespread belief that these conditions are primarily musculoskeletal in nature, rather than multisystem disorders.

## METHODS

### Study design: nationwide electronic cohort study

Anonymised record linkage and hosting is carried out in the Secure Anonymised Information Linkage (SAIL) databank[26] on routinely collected data held in health and social care datasets. All data within the SAIL gateway are treated in accordance with the Data Protection Act 2018 and complies with the EU General Data Protection Regulation (GDPR).

We used data from a variety of datasets between 1 July 1990 and 30 June 2017 to create the anonymised e-cohort and case–control studies. The primary care data cover about 80% of all coded information held by general practitioners (GPs) in Wales. The Welsh Demographics Service (WDS) contains key statistics, such as gender, week of birth, date of death and practice migration status for everyone in Wales registered with a GP. The Patient Episode Database for Wales (PEDW) contains all inpatient hospital admissions to a Welsh hospital. WDS and PEDW data are available for the whole of Wales. The SAIL databank enables the anonymised matching of individuals across these different datasets using a person level anonymised linkage field (ALF).[26]

### Cohort preparation

We identified Welsh residents with a Read Version 2[27] diagnostic code of EDS or JHS in primary care data or International Statistical Classification of Diseases and Related Health Problems, 10th Revision (ICD-10) diagnostic codes[28] in secondary care data (hospital admissions) between 1 July 1990 (or the start of the dataset if later) and 30 June 2017. This date marks the end of maximum data coverage across all datasets. The EDS subclassification in Read Version 2 contains some, but not all, of the subtypes which were in use prior to 1997 and as a result, the reliability of any subtype data must be highly

**Table 1** Clinical coding for EDS and JHS

| Read code descriptions (based on pre-1997 nomenclature) | EDS type according to the Villefranche criteria | EDS type according to the March 2017 criteria | Read code version 2 | ICD-10 code |
|---|---|---|---|---|
| Ehlers-Danlos syndrome | | | PGy2. | Q79.6 |
| Ehlers-Danlos syndrome type I | Classical type | Classical EDS | PGy20 | |
| Ehlers-Danlos syndrome type II | | | PGy21 | |
| Ehlers-Danlos syndrome type III | Hypermobility type | Hypermobile EDS or hypermobility spectrum disorder | PGy22 | |
| Ehlers-Danlos syndrome type IV | Vascular type | Vascular EDS | PGy23 | |
| Ehlers-Danlos syndrome type V | X-linked type | No longer classified as EDS | PGy24 | |
| Ehlers-Danlos syndrome type VI | Kyphoscoliotic type | Kyphoscoliotic EDS | PGy25 | |
| Ehlers-Danlos syndrome type VII | Arthrochalasia type Dermatosparaxis type | Arthrochalasia EDS Dermatosparaxis EDS | PGy26 | |
| Ehlers-Danlos syndrome type VIII | Periodontitis type | Periodontal EDS | PGy27 | |
| Hypermobility syndrome (JHS according to the Brighton Criteria) | Hypermobility type | Hypermobile EDS or hypermobility spectrum disorder | N235. | 728.5 |

EDS, Ehlers-Danlos syndrome; ICD-10, International Statistical Classification of Diseases and Related Health Problems, 10th Revision; JHS, joint hypermobility syndrome.

questionable (see table 1). Due to the lack of available correct subcodes for EDS subtypes, the fact that the overwhelming majority of patients simply had the header code (86% of those coded as EDS, with a further 12% coded as hypermobile EDS (hEDS)), and that other EDS types are genuinely rare, all codes for EDS were combined. ICD-10 codes do not distinguish between any subtypes of EDS (see table 1). Only ALFs with good matching status were included in the study, that is, direct match on either National Health Service (NHS) number or on surname, first name, postcode, date of birth and gender; or fuzzy matching with a probability of ≥90%.

We created one dataset for diagnoses in the GP data and another for diagnoses in the hospital data. Both datasets were linked to the week of birth, gender and date of death information in WDS on their ALF and then combined to create a cohort of people with EDS/JHS in either GP or hospital data, identifying any duplications and keeping the earliest diagnosis date for any individual appearing in both datasets.

### Analysis

Data linkage and data preparation within the SAIL databank were conducted using IBM DB2 10.5 SQL. Data were then imported into R (Version 3.4.1),[29] which was used for all statistical analyses. The mean age at first diagnosis between male and female subjects was compared and CIs of the difference calculated.

The denominator of the diagnosed prevalence and incidence of EDS and JHS in secondary care was calculated based on the total number of individuals with recorded gender, registered and living in Wales between 01 July

1990 and 30 June 2017 for each full year of the study, respectively. The prevalence and incidence in primary care denominator was further adjusted to include only people living in Wales and whose GP practice was contributing data to SAIL. The prevalence and incidence in primary and secondary care was then added together to create an overall estimate of the prevalence and incidence in Wales.

### Case–control comparison

A nested case–control method was used. Each case was matched to four controls with the same gender and similar age profiles (within 45 days of the week of birth). We implemented strict criteria for selection to the case–control cohort. Both cases and controls had to (1) have uninterrupted GP registrations for 1 year before and 1 year after the date of the relevant diagnosis (or died during follow-up); (2) be registered with a GP submitting data to SAIL either at the matching date or afterwards; (3) have been registered with a GP that consistently recorded data across their patient profile. The latter avoids diagnoses that were retrospectively entered for a time period when the GP practice did not fully implement the use of electronic records (less than 10% of the data they recorded during 2009). Although this reduced the number of cases and controls we were able to analyse, it avoids data quality bias, especially during the early years of this study, when GPs were converting to the use of computer systems and databases. Controls with any type of diagnosed hereditary connective tissue disorder were excluded. Preliminary analysis of the combined cohort indicated that adjustment for deprivation was not necessary (ie, equal

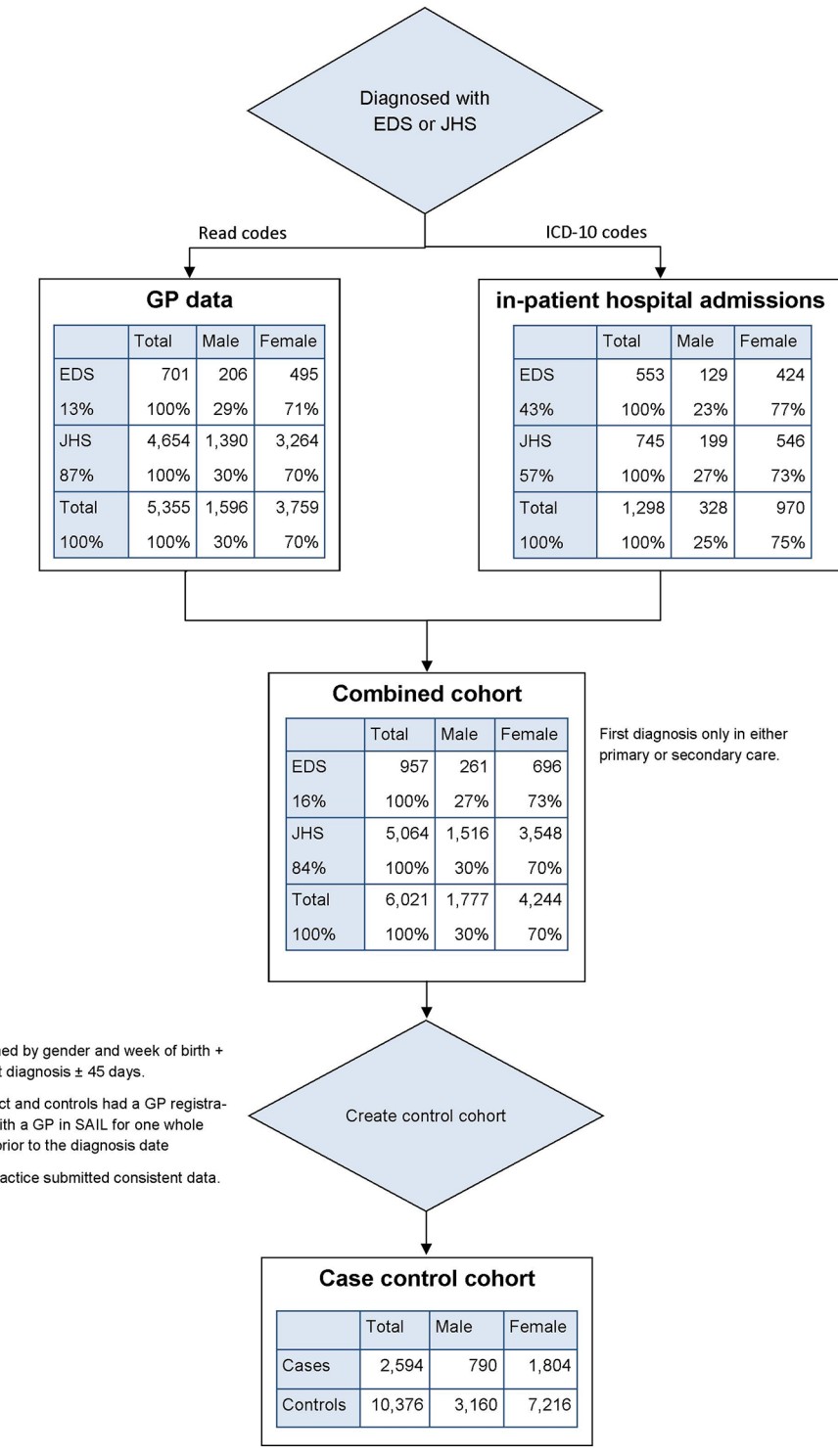

**Figure 1** Flow diagram of EDS/JHS cohort and case–control cohort creation. EDS, Ehlers-Danlos syndrome; GP, general practitioner; ICD-10, International Statistical Classification of Diseases and Related Health Problems, 10th Revision; JHS, joint hypermobility syndrome; SAIL, Secure Anonymised Information Linkage

distribution of people across deprivation quintiles). We then calculated ORs between cases and controls using Read chapters (excluding the Read codes for EDS and JHS). This method counts the number of people with a code in each category; multiple codes for the same person in the same category are therefore not included.

All results that affected at least 5 cases or 20 controls were visualised using forest plots.

### Ethical approval
The study design uses anonymised data and therefore the need for ethical approval and participant consent was

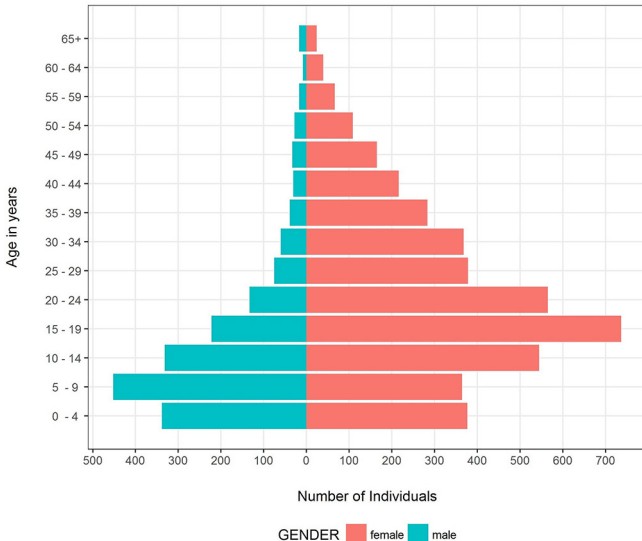

**Figure 2** Age at first coded diagnosis of EDS/JHS by age group and gender. EDS, Ehlers-Danlos syndrome; JHS, joint hypermobility syndrome.

waived by the approving Institutional Review Board, the UK National Health Service Research Ethics Committee. The SAIL independent Information Governance Review Panel approved the study.

## Patient and public involvement

Two of the authors of this paper have been diagnosed with symptomatic joint hypermobility disorders. This study used routinely collected data, and we were not able to involve members of the public but will be disseminating our findings widely, including directly to patients via social media and through our links with patient organisations.

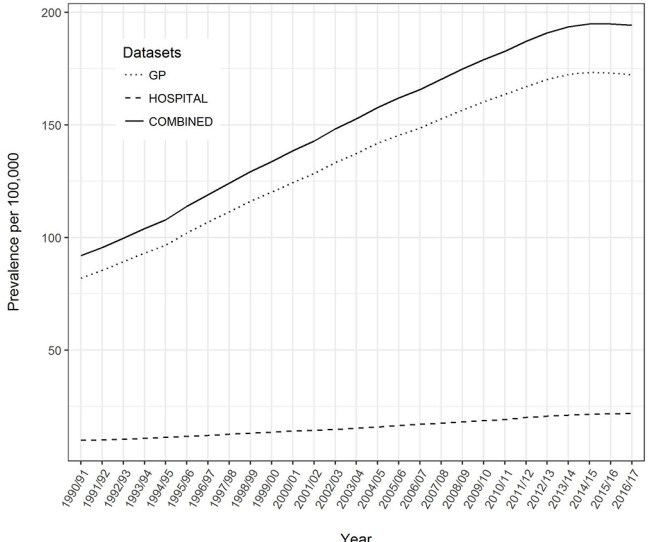

**Figure 3** Prevalence of coded diagnosis of JHS/EDS in primary care, hospital inpatient and combined over time. EDS, Ehlers-Danlos syndrome; GP, general practitioner; JHS, joint hypermobility syndrome.

# RESULTS

## EDS/JHS in primary care data

5355 individuals with a diagnosis of either EDS or JHS with valid birth and gender information were identified. Of these, 4654 (87%) had a diagnosis of JHS and 701 (13%) of EDS. The Read code for the EDS subtype was only used for 136 (19%) individuals with 114 (16%) identified as EDS-hypermobility type and 22 (3%) as other EDS subtypes. 3759 (70%) of those diagnosed with EDS/JHS were women (see figure 1).

## EDS/JHS in hospital data

A total of 1298 individuals were found in the hospital data of whom 970 (75%) were women: 745 (57%) had a diagnosis of JHS and 553 (43%) EDS (see figure 1).

## Demographics of combined EDS/JHS cohort

5355 (89%) of the cases could be found in the primary care data with the remainder in the hospital cohort. Combining the results from primary and secondary care led to a cohort of 6021 distinct individuals. 5064 (84%) were coded with JHS and 957 (16%) with EDS. 4244 (70%) of patients were women. The age at first diagnosis peaked in the age group 5–9 years for men and 15–19 years for women (see figure 2). There was a significant difference of 8.5 years in the mean age of diagnosis between men and women (95% CI: 7.70 to 9.22): 9.6 years in EDS (95% CI: 6.85 to 12.31) and 8.3 years in JHS (95% CI: 7.58 to 9.11). 72% of men were diagnosed during childhood (age <18 years) in contrast to only 41% of women.

2016/2017 is the latest year for which we have complete data and could therefore derive prevalence. During this year, 2 668 902 people were registered with a GP in Wales submitting data to SAIL, of whom 4598 had a diagnostic code of EDS/JHS which first appeared in the primary care data (172 in 100 000). A further 711 people out of the 3 239 153 registered with any GP in Wales during 2016/2017 had an EDS/JHS diagnosis which first appears in secondary care data (22 in 100 000). There is an increasing rate of coded diagnoses throughout the period. Assuming that the GP data are representative of the whole of Wales this leads to a combined point prevalence of 194 in 100 000 at the end of the study period. This corresponds to about 10 cases in a practice of 5000 patients (see figure 3). The incidence of EDS/JHS over this time period is shown in online supplementary figure 1.

## Factors associated with JHS/EDS

2597 cases had good GP data coverage at the age of diagnosis and could be matched by age and gender with controls (see figure 1). 1340 cases (men: 561; women: 779) were first diagnosed before the age of 18 years and 1254 cases (men: 229; women: 1025) above this age. The people in the nested case–control cohort were slightly older than the overall cohort (data not shown here).

Looking at the time frame of 1 year on either side of the first coded diagnosis of EDS/JHS among young people

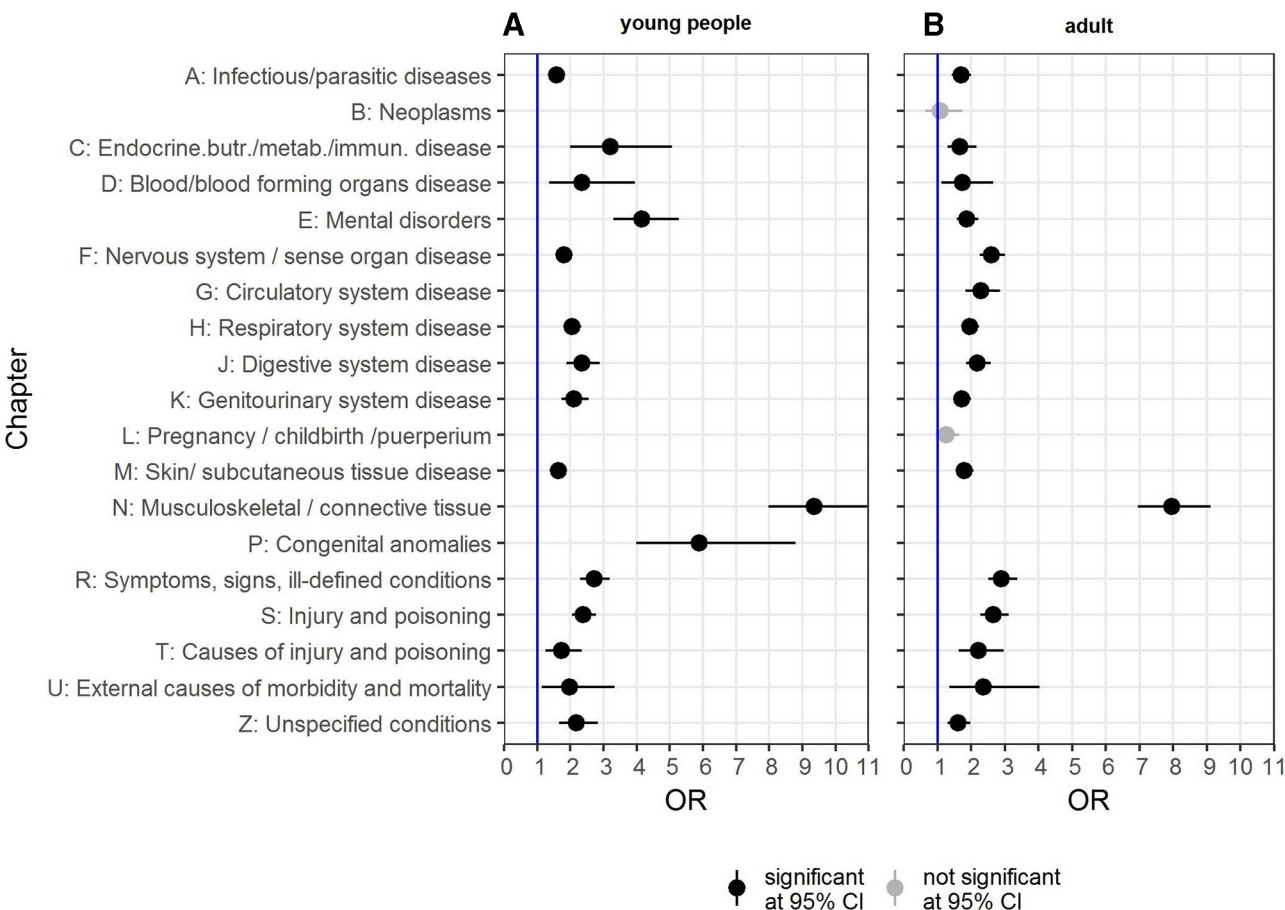

**Figure 4** ORs of Read chapter diagnoses for (A) young people (<18 years of age) and (B) adults (≥18 years of age) within 12 months before and after EDS/JHS diagnosis. Presented are all results that affect at least 5 cases or 20 controls (perinatal conditions, Chapter Q, are not shown as neither young people nor adults had the required minimum number of cases/controls). EDS, Ehlers-Danlos syndrome; JHS, joint hypermobility syndrome.

(age <18 years), there were significantly more additional diagnoses in 16 out of 20 Read code disease categories compared with their controls (see figure 4A). The top three Read diagnosis chapters with increased odds for the EDS/JHS cohort were for musculoskeletal conditions (OR 9.36, 95% CI: 7.98 to 11.00), congenital anomalies (OR 5.89; 95% CI: 3.98 to 8.80) and mental disorders (OR 4.16; 95% CI: 3.29 to 5.27).

People who were diagnosed as adults (age ≥18 years) had also significantly more diagnoses in 16 out of 20 Read code categories than their controls (see figure 4B). The top three Read diagnosis chapters for adults with higher odds in the EDS/JHS cohort were musculoskeletal disorders (OR 7.95; 95% CI: 6.95 to 9.12), congenital anomalies (OR 5.18; 95% CI: 2.78 to 9.78) and symptoms, signs and ill-defined conditions (OR 2.9; 95% CI: 2.5 to 3.37). Circulatory system disease (OR 2.29; 95% CI: 1.83 to 2.86) and mental disorders remained significant (OR 1.87; 95% CI: 1.57 to 2.22), but not to the same extend as they were for young people.

Young people showed significantly higher odds for prescriptions in 14 out of 17 Read code categories then their controls (see figure 5A). The top three prescriptions of Read chapters with increased odds for the EDS/

JHS cohort were for (1) musculoskeletal drugs (OR 3.65; 95% CI: 3.18 to 4.18), (2) gastrointestinal drugs (OR 3.02; 95% CI: 2.54 to 3.58) and (3) haematology/dietetic drugs (OR 2.54; 95% CI: 2.06 to 3.11).

Adults had significantly higher odds of prescriptions for 15 out of 17 Read code categories (see figure 5B). The top three prescriptions with higher odds for EDS/ JHS people were for (1) musculoskeletal drugs (OR 5.17; 95% CI: 4.53 to 5.9), (2) central nervous system drugs (OR 3.9; 95% CI: 3.41 to 4.46) and (3) chemotherapy/ immunosuppressant drugs (OR 3.03; 95% CI: 1.89 to 4.8). Gastrointestinal drugs (OR 2.85; 95% CI: 2.5 to 3.24) and haematology/dietetic drugs (OR 2.21; 95% CI: 1.9 to 2.57) remain significant, but at slightly lower levels than in the young EDS/JHS population.

## DISCUSSION

This work examined the epidemiology of EDS and JHS and found a combined diagnosed prevalence of 194.2 per 100 000 (0.19%) or 1 in 500 people in Wales; hEDS or HSD within the 2017 classification. We found a steadily increasing rate of diagnosis over the past 27 years (see online supplementary figure 1), as well as higher rates of

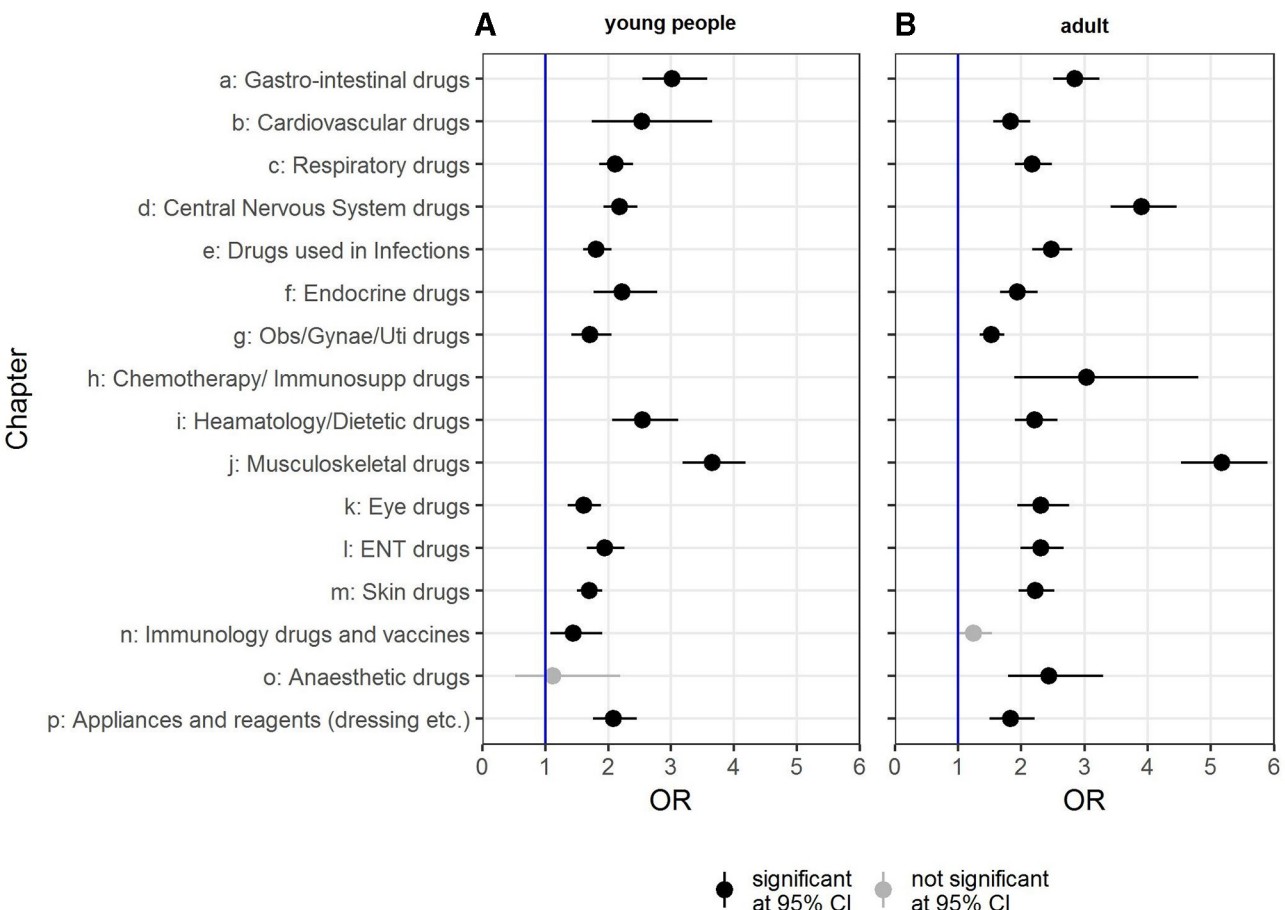

**Figure 5** ORs of Read chapter prescriptions for young people (<18 years of age) and adults (≥18 years of age) within 12 months before and after EDS/JHS diagnosis. Presented are all results that affect at least 5 cases or 20 controls (incontinence and stoma appliances, Chapters Q and S, are not shown as neither young people nor adults had the required minimum number of cases/results). EDS, Ehlers-Danlos syndrome; JHS, joint hypermobility syndrome.

diagnoses for other conditions and prescriptions within 12 months (before and after) of the recorded first diagnosis in most categories. This suggests that hEDS/HSD, when considered together, do not meet the definition of rare conditions[23] and have widespread effects across multiple body systems.

It is well-known that EDS is poorly recognised in children[30 31] and initial symptoms and EDS-associated diagnoses can appear to be simply a 'normal' pattern of childhood illness when taken as an isolated event. Furthermore, children with hEDS often present with symptoms that can lead to a misdiagnosis of mental illness or consideration of child abuse.[12 32] Suspicion of abuse has been shown to be extremely damaging to the mental health of the parent(s) and can lead to an avoidance of accessing healthcare or other public services, such as schools.[33] The prolonged and sometimes traumatic diagnosis and/or misdiagnosis process in EDS can lead to further disengagement with services.[34] The lack of a timely diagnosis has great implications for disease management and progression and impedes the appropriate consideration of surgical interventions[7 35–38] as well as pregnancy and birth planning.[17] It is perhaps only in stepping back to look at the pattern of effects across multiple body systems

that practitioners might begin to consider a connective tissue disorder.

### Strengths and limitations

The strength of this study is that we were able to combine diagnostic codes from several primary and secondary healthcare providers to create a large cohort of individuals with EDS/JHS. We have 27 years of data with at least 11 years of very good data coverage in the key datasets, which further improves with each data update of the SAIL databank; however, data coverage for the first couple of years is less comprehensive.

The majority of subjects were identified via their primary care data, which is a strength and a weakness. As 89% of cases were identified through primary care, data studies not using primary care data may underestimate the prevalence of hEDS/HSD. We are unable to quantify how many people are suffering from hEDS or HSD but remain undiagnosed. However, we cannot comment on the reliability of the diagnoses in the primary care dataset. It is also likely that the majority of cases were not actually diagnosed in primary care, but their entries were created through secondary care contacts, such as outpatient

appointments or musculoskeletal assessment clinics, but coded data are lacking from these sources.

Although a snapshot of Read chapters codes that are more prevalent in our JHS/EDS cohort does not allow us to look at specific diagnoses and prescriptions, they can all be matched to conditions associated with EDS/JHS in the literature, for instance pain, fatigue, cardiovascular, gastrointestinal and gynaecological disorders, dysautonomia, mast cell activation as well as urinary tract infections.[7] It needs to be stressed that these results exclude codes for EDS/JHS and that these are not part of the results for congenital anomalies or musculoskeletal conditions. We hope in future work to examine in greater detail these findings of significant differences between people with hEDS/HSD and others in order that we can better understand the nature of this condition, as well as potentially improving diagnostic recognition. For instance, immunosuppressant drugs are in the same Read chapter as chemotherapy drugs. Higher use of these prescriptions in the hEDS/HSD cohort could plausibly be linked to the known increased comorbid existence of disorders such as inflammatory bowel disease, inflammatory arthropathies, systemic lupus erythematosus and other autoimmune conditions,[39] and is less likely to be due to a higher rate of use of chemotherapeutic agents.

We conclude that EDS/HSD are not rare conditions and are associated with significantly increased odds of additional diagnoses and use of medications across many body systems. There is a large gender difference in the age of diagnosis, with many women not diagnosed until adulthood. Early diagnosis, however, is crucial to patients, the administration of preventive therapies, the investigation of comorbid conditions and the overall management process. Further research is needed to understand patient pathways, comorbidities and progression of associated symptoms and diseases. Health services should be aware of these findings for the provision of training, diagnostic and treatment services for the many tens of thousands of patients living with these life-changing conditions throughout the UK and beyond.

**Contributors** JCD conceived the project. STB and MDA contributed to the study design and analysis plan. RAL validated clinical codes in primary care. EC validated clinical codes in secondary care. JCD undertook the analysis. JCD and EJR carried out the literature reviews and drafted the manuscript. All authors reviewed the manuscript and approved the final version for submission.

**Funding** This work was supported by The Farr Institute. The Farr Institute was supported by a 10-funder consortium: Arthritis Research UK, the British Heart Foundation, Cancer Research UK, the Economic and Social Research Council, the Engineering and Physical Sciences Research Council, the Medical Research Council, the National Institute of Health Research, the National Institute for Social Care and Health Research (Welsh Assembly Government), the Chief Scientist Office (Scottish Government Health Directorates) and the Wellcome Trust (MRC Grant No: MR/K006525/1). This work was also supported by Health Data Research UK which receives its funding from HDR UK Ltd (NIWA1) funded by the UK Medical Research Council, Engineering and Physical Sciences Research Council, Economic and Social Research Council, Department of Health and Social Care (England), Chief Scientist Office of the Scottish Government Health and Social Care Directorates, Health and Social Care Research and Development Division (Welsh Government), Public Health Agency (Northern Ireland), British Heart Foundation (BHF) and the Wellcome Trust.

The above funders played no role in the study design, in the collection, analysis, or interpretation of data; in the writing of the report; or in the decision to submit the article for publication. The researchers are independent from the funders and all authors, external and internal, had full access to all of the data in the study and take responsibility for the integrity of the data and the accuracy of the data analysis.

**Competing interests** None declared.

**Patient consent for publication** Not required.

**Ethics approval** The SAIL independent Information Governance Review Panel approved the study.

**Provenance and peer review** Not commissioned; externally peer reviewed.

**Data availability statement** Data may be obtained from a third party and are not publicly available.

**Author note** The authors are planning to disseminate our results to patient groups using social media.

**ORCID iDs**
Joanne C Demmler http://orcid.org/0000-0002-4675-6271
Ronan A Lyons http://orcid.org/0000-0001-5225-000X

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
