## [Reviewer comments · BMJ Open]

ARTICLE DETAILS

TITLE (PROVISIONAL)	Diagnosed prevalence of Ehlers-Danlos syndrome and Hypermobility Spectrum Disorder in Wales, UK: a national electronic cohort study and case-control comparison
AUTHORS	Demmler, Joanne; Atkinson, Mark; Reinhold, Emma; Choy, Ernest; Lyons, Ronan; Brophy, Sinead

VERSION 1 – REVIEW

REVIEWER	Jacobs JWG UMCU, Utrecht, the Netherlands
REVIEW RETURNED	05-Jun-2019

GENERAL COMMENTS	In this study using Welsh data from primary care and hospital admissions routinely collected from July 1, 1990 to June 30, 2017, among all individuals registered, individuals with a diagnostic code of Ehlers-Danlos syndrome (EDS, no specification of type EDS) or joint hypermobility syndrome (JHS) were identified. These were lumped as "EDS or JHS" and as "EDS or hypermobility spectrum disorder (HSD)" as alternative term, and the prevalence in Wales was calculated, concluding that these are not rare disorders. Each case was matched for gender and age with 4 controls and the frequencies and character of additional diagnoses and prescriptions routinely collected were compared between cases and controls, showing more additional diagnoses and prescriptions in the cases, concluding that "EDS or JHS" or "EDS or HSD" affect more body parts and organs than the musculoskeletal system and soft tissues. 1) a drawback is the imprecision of coded diagnoses, with no discrimination of EDS types. The drawback of possible lack of reliability of coded diagnoses is recognised by the authors in the bulleted limitations. Although indeed hypermobile EDS (hEDS) is the most frequent type, in hospital admission diagnosis codes, vascular EDS might be relatively frequently occurring, because hEDS is a diagnosis not frequently leading to admission, while vEDS is: vEDS is a life-threatening disorder with severe complications, while hEDS is not. This questions the validity of lumping hospital and primary care data. Data on prevalence of vEDS would have been very interesting. These issues should be discussed. 2) although one could argue "EDS or JHS" and "EDS or HSD" comprise the same individuals, it should be recognised that an individual who was classified as EDS hypermobility type according the Villefranche criteria might well not meet the 2017 hEDS criteria; this should be discussed. 3) because of issue 1) en 2), I cannot agree with the text of the aim in the abstract; "To describe the epidemiology of diagnosed
--

	Hypermobility Spectrum Disorder (HSD) and Ehlers-Danlos Syndromes (EDS) under the 2017 classification": it is not "under the 2017 classification" 4) the authors speak of "additional diagnoses", but these might be additional codings of the same diagnosis. For instance, in Figure 4, I wonder whether the high odds of congenital anomalies are additional codings of the congenital hypermobility syndrome; likewise, the category "musculoskeletal/connective tissue" with its high odds in cases seems additional codings of (consequences of) the hypermobility syndrome. This should be discussed as it is now an essential part of the conclusion. 5) while it seems predictable that indeed cases used more "Musculoskeletal drugs" (Figure 5), and "Gastro-intestinal drugs" (co-medication of NSAID?), the higher use of "Chemotherapy/Immunosupp drugs" is fully surprising to me and should be explained. Moreover, what is the rationale of the lumping of the "Haematology/Dietetic" category?, what kind of drugs are these? 6) It seems that all controls were derived from GP data only? It is known that patients who have been admitted use more medications than those not admitted, irrespective of a diagnosis. Could this explain the higher frequency of use of medications in cases, who also are derived from hospital data? Should cases not have been matched, not only for gender and age as has been done, but also for source of data/database? Is there a case for reanalysing these data? 7) minor: a) "● Connective tissue disorders are recognised as affecting more than the musculoskeletal system." should better read ● Congenital connective tissue disorders are recognised as affecting more than the musculoskeletal system.". Although the message would also apply acquired, immunologic connective tissue disorders, the authors did not investigate this. b) The focus: "To provide physicians with an overview of the epidemiology of Ehlers-Danlos Syndromes; prevalence and co-morbidities." could raise the false expectation that the epidemiology of all Ehlers-Danlos syndromes/types is the focus, not only hEDS.
--	---

REVIEWER	M.C. Scheper University of Applied Sciences Rotterdam, Institute for Health Innovation, Physical therapy, Rotterdam, the Netherlands The Hague university for Applied Sciences, faculty of health, nutrition and sports, medical health technology and innovation, the Hague, the Netherlands
REVIEW RETURNED	20-Jun-2019

GENERAL COMMENTS	The manuscript is of great importance to the field and addresses issues that are vital for further progression of scientific advances in this field as well as clinical and societal recognition. In general the manuscript is straight forward and well written, however I have some comments concerning the methodology and statistics. Methodology:
---

	Please provide a clearer understanding of the validity of the used chapters of the GP toolkit. When looking at the literature in this field and the different chapters as presented in the outcomes, it can be argued that these some of these outcomes overlap. For instance multi-systemic involvement has been shown to be the greatest contributor to disability in children with EDS which can contain issues in orthostatic intolerance, incontinence, Diarrhoea etc. In addition central nervous system sensitization in terms of generalized hyperalgesia is highly prevalent in EDS (both adults and children). From a clinical GP perspective it makes sense to consider these issues separately, however from a methodological perspective, increasing the number of potential outcomes that in theory can have a high probability to be correlated may lead to false associations. Even in this large cohort the risk still remains. Please provide an argument for retaining these categories (other than clinical) by excluding the risk of co-linearity (statistical assumptions) or combine these subgroups. Considering the potential effect of age and gender on the OR ratio's was a confounder analysis performed? Splitting the data on age can be beneficial but may also lead to further fragmentation of variance and add to the issues addressed above. Correcting for age and gender in the analysis as a covariable could limit the influence of bias and retaining the natural variance of the dataset. Please elaborate on this choice. The discussion is an insider on the field very clear but I would suggest to elaborate a bit more on the potential issues that makes early detection so difficult in children.
--	---

VERSION 1 – AUTHOR RESPONSE

Reviewer: 1

Reviewer Name: Jacobs JWG

Institution and Country: UMCU, Utrecht, the Netherlands Please state any competing interests or state 'None declared': none

Please leave your comments for the authors below In this study using Welsh data from primary care and hospital admissions routinely collected from July 1, 1990 to June 30, 2017, among all individuals registered, individuals with a diagnostic code of Ehlers-Danlos syndrome (EDS, no specification of type EDS) or joint hypermobility syndrome (JHS) were identified. These were lumped as "EDS or JHS" and as "EDS or hypermobility spectrum disorder (HSD)" as alternative term, and the prevalence in Wales was calculated, concluding that these are not rare disorders. Each case was matched for gender and age with 4 controls and the frequencies and character of additional diagnoses and prescriptions routinely collected were compared between cases and controls, showing more additional diagnoses and prescriptions in the cases, concluding that "EDS or JHS" or "EDS or HSD" affect more body parts and organs than the musculoskeletal system and soft tissues.

1) a drawback is the imprecision of coded diagnoses, with no discrimination of EDS types. The drawback of possible lack of reliability of coded diagnoses is recognised by the authors in the bulleted limitations. Although indeed hypermobile EDS (hEDS) is the most frequent type, in hospital admission diagnosis codes, vascular EDS might be relatively frequently occurring, because hEDS is a diagnosis

not frequently leading to admission, while vEDS is: vEDS is a life-threatening disorder with severe complications, while hEDS is not. This questions the validity of lumping hospital and primary care data. Data on prevalence of vEDS would have been very interesting. These issues should be discussed.

Thank you for your comment. Yes, indeed, we acknowledge there is imprecision in the coding. In primary care, header codes are often used until a more precise diagnosis subtype can be ascertained or for speed. With respect to the available sub-codes for EDS in primary care, these were outdated and not reflective of even the pre-2017 Villefranche terminology. However, in contrast to hEDS, vEDS is truly rare, with a prevalence of between 1 in 50,000 and 1 in 200,000. There are approximately 250 patients registered on the Annabelle's Challenge (vEDS) patient database throughout the UK. Less than 5 cases were explicitly coded as vEDS out of 5,355 cases found in the GP data. Furthermore cases would almost certainly be known to their GP, due to the significantly different presentation, risk and treatment approaches in vEDS compared with hEDS. It is therefore highly unlikely that cases of vEDS had any significant effect on the data, particularly when you consider that only 16% of the whole cohort had an EDS code with 84% coded as JHS. Due to the lack of available correct sub-codes for EDS subtypes, the fact that the overwhelming majority of patients simply had the header code, and our clinical experience that all types other than EDS are genuinely rare (Dr Alan Hakim sends only around 2% of patients referred to him on for genetic testing for other subtypes, with the remainder receiving a clinical diagnosis of hEDS/HSD – personal communication) we opted to combine all patients with either the EDS header code or any sub-code. As we point out in the text, 89% of the cases could be identified from the primary care data.

We have clarified this in the text: "Due to the lack of available correct sub-codes for EDS subtypes, the fact that the overwhelming majority of patients simply had the header code (86% of those coded as EDS, with a further 12% coded as hEDS), and that other EDS types are genuinely rare, all codes for EDS were combined."

2) although one could argue "EDS or JHS" and "EDS or HSD" comprise the same individuals, it should be recognised that an individual who was classified as EDS hypermobility type according to the Villefranche criteria might well not meet the 2017 hEDS criteria; this should be discussed.

Thank you for your comment. We have outlined the differences between the diagnostic criteria in the introduction, table 1 and in textbox 1

3) because of issue 1) and 2), I cannot agree with the text of the aim in the abstract; "To describe the epidemiology of diagnosed Hypermobility Spectrum Disorder (HSD) and Ehlers-Danlos Syndromes (EDS) under the 2017 classification": it is not "under the 2017 classification"

Thank you for your comment. When the 2017 classification was published, it was made clear that no 'old' diagnosis was to be removed, but that those who might previously be diagnosed as either JHS or EDS-HT, if reassessed now (e.g. for research purposes) would fall under the hEDS or HSD criteria. However, we have removed this from the abstract as it might be confusing to the reader.

4) the authors speak of "additional diagnoses", but these might be additional codings of the same diagnosis. For instance, in Figure 4, I wonder whether the high odds of congenital anomalies are additional codings of the congenital hypermobility syndrome; likewise, the category "musculoskeletal/connective tissue" with its high odds in cases seems additional codings of (consequences of) the hypermobility syndrome. This should be discussed as it is now an essential part of the conclusion.

Thank you for your comment. These are coded diagnoses excluding the diagnosis codes for JHS and EDS (see methods). Also, we counted people with these codes and not the numbers of codes themselves, therefore only one code in this category was selected for each person. Furthermore, the hypermobility syndromes do not come under the read-code sub-heading of congenital anomalies, so it is unlikely to be the explanation of this apparent difference. Some of these codes will be a direct reflection of comorbidities associated with EDS/JHS, but this is exactly the point we wanted to make and is part of our discussion.

The text now reads: "We then calculated odds ratios between cases and controls using Read chapters (excluding the Read codes for EDS and JHS)."

5) while it seems predictable that indeed cases used more "Musculoskeletal drugs" (Figure 5), and "Gastro-intestinal drugs" (co-medication of NSAID?), the higher use of "Chemotherapy/Immunosupp drugs" is fully surprising to me and should be explained. Moreover, what is the rationale of the lumping of the "Haematology/Dietetic" category?, what kind of drugs are these?

Thank you for your comment. We believe this might be connected to the use of azathioprine/ciclosporin/methotrexate for co-existent Crohns/colitis or rheumatoid disease, plus a few 'imab' drugs for auto-immune diseases such as lupus, or for rheumatoid arthritis, and some MS drugs (MS is thought to have a higher prevalence in this population) However, in this paper we do not seek to explain, only to describe in a broad sense the differences between the categories. We hope to undertake further work to dig down into the detail, and the reasons behind these findings, but we feel it would be premature to attempt to 'explain' our findings at this stage.

Haematology/dietetic drugs are both included in chapter 9 of the BNF, the model we used for our analysis. This is why they are shown together. They are mainly supplements, including iron, folic acid, vitamin B12, calcium and magnesium.]

The text now reads: "We hope in future work to examine in greater detail these findings of significant differences between people with hEDS/HSD and others in order that we can better understand the nature of this condition, as well as potentially improving diagnostic recognition. Having created this case-control cohort, further examination is made simpler as this first step has already been made."

6) It seems that all controls were derived from GP data only? It is known that patients who have been admitted use more medications than those not admitted, irrespective of a diagnosis. Could this explain the higher frequency of use of medications in cases, who also are derived from hospital data? Should cases not have been matched, not only for gender and age as has been done, but also for source of data/database? Is there a case for reanalysing these data?

Thank you for your comment. All cases and controls were matched based on their quality in the GP data, but some of them might only have had a diagnosis code in the hospital data. However, both cases and controls might have had hospital admissions. 6% of all cases were only to be found in the hospital data (with 19% being identifiable in both hospital and GP data). Yes, it is correct that these people's medication might be connected to a higher level of hospital admissions, however, we wanted to show how they appear within the Primary Care record which might help in the earlier diagnosis of patients.

7) minor: a) "• Connective tissue disorders are recognised as affecting more than the musculoskeletal system." should better read • Congenital connective tissue disorders are recognised as affecting more than the musculoskeletal system.". Although the message would also apply acquired, immunologic connective tissue disorders, the authors did not investigate this.

b) The focus: "To provide physicians with an overview of the epidemiology of Ehlers-Danlos Syndromes; prevalence and co-morbidities." could raise the false expectation that the epidemiology of all Ehlers-Danlos syndromes/types is the focus, not only hEDS.

Thank you for your comment. The article focus has been removed by the editor.

Reviewer: 2

Reviewer Name: M.C. Scheper

Institution and Country: University of Applied Sciences Rotterdam, Institute for Health Innovation, Physical therapy, Rotterdam, the Netherlands The Hague university for Applied Sciences, faculty of health, nutrition and sports, medical health technology and innovation, the Hague, the Netherlands
Please state any competing interests or state 'None declared': None declared

Please leave your comments for the authors below The manuscript is of great importance to the field and addresses issues that are vital for further progression of scientific advances in this field as well as clinical and societal recognition.

In general the manuscript is straight forward and well written, however I have some comments concerning the methodology and statistics.

Methodology:

Please provide a clearer understanding of the validity of the used chapters of the GP toolkit. When looking at the literature in this field and the different chapters as presented in the outcomes, it can be argued that these some of these outcomes overlap. For instance multi-systemic involvement has been shown to be the greatest contributor to disability in children with EDS which can contain issues in orthostatic intolerance, incontinence, Diarrhoea etc. In addition central nervous system sensitization in terms of generalized hyperalgesia is highly prevalent in EDS (both adults and children). From a clinical GP perspective it makes sense to consider these issues separately, however from a methodological perspective, increasing the number of potential outcomes that in theory can have a high probability to be correlated may lead to false associations. Even in this large cohort the risk still remains. Please provide an argument for retaining these categories (other than clinical) by excluding the risk of co-linearity (statistical assumptions) or combine these subgroups.

Thank you for your comment. Read version 2 is a categorical coding system (not related to the EDS toolkit). We used this categorisation as a step to point out areas of interest/concern in EDS patients. This study did not perform statistical modelling on an outcome variable, but calculated odds ratios.

Considering the potential effect of age and gender on the OR ratio's was a confounder analysis performed? Splitting the data on age can be beneficial but may also lead to further fragmentation of variance and add to the issues addressed above. Correcting for age and gender in the analysis as a covariable could limit the influence of bias and retaining the natural variance of the dataset. Please elaborate on this choice.

Thank you for your comment. We have presented a split of children versus adults instead of performing a detailed confounder analysis. Splitting the data further by age and gender was not possible as several age classes would not have enough cases for meaningful statistical results and/or

would be below the number of records that can be released by the SAIL data guardian. We hope that splitting by age and gender will be feasible in a future data extract, at the time of writing the number of cases and controls were very restricted by the GP data coverage .

The decision to split the data into paediatric and adult cohorts was considered carefully by our team. In the end, we concluded that it was important to demonstrate that significant multi-systemic effects can occur even in childhood; something which is important for paediatric practice in particular, where historically there has been less of an awareness of such findings. If we had combined all ages together, paediatricians would not be able to see how their patients are impacted, and might have concluded that these messages were not relevant to children. This is of particular importance at a time when some paediatric practitioners are suggesting that a child presenting with symptoms in more than one body system should be considered as being at risk of fabricated or induced illness. Given our findings, we felt that presenting the data separately may prove important in suggesting that disorders such as EDS must also be considered alongside child abuse as a potential explanation in such situations. We did exclude sections where the numbers became low to avoid introducing bias. We would also argue that by presenting the adult dataset alongside the paediatric, we are beginning to demonstrate that these findings are replicable across different population groups.

The discussion is an insider on the field very clear but I would suggest to elaborate a bit more on the potential issues that makes early detection so difficult in children.

Initial symptoms and EDS-associated diagnoses can appear to be simply a 'normal' pattern of childhood illness when each is taken as an isolated event. It is perhaps only in stepping back to look at the pattern of effects across multiple body systems that practitioners might begin to consider a connective tissue disorder.

The text now reads: "...and initial symptoms and EDS-associated diagnoses can appear to be simply a 'normal' pattern of childhood illness when each is taken as an isolated event. ... It is perhaps only in stepping back to look at the pattern of effects across multiple body systems that practitioners might begin to consider a connective tissue disorder."

VERSION 2 – REVIEW

REVIEWER	Jacobs JWG UMCU, Netherlands
REVIEW RETURNED	08-Aug-2019

GENERAL COMMENTS	The issues raised are in general addressed adequately. However, the authors should try to convince the readers of their paper more that their key findings of more additional codings and medication use are real, not biased, as these findings are used to show that EDS is associated with comorbidity. This can be done in two ways. 1) I would suggest that the authors add the additional explanation given in their rebuttal on my previous issue 4 also to the paper, e.g. in the methods section. 2) My previous issue 5 was that the higher use of "Chemotherapy/Immunosupp drugs" is fully surprising to me and should be explained. The authors answer that they do not seek to explain, only to describe in a broad sense the differences between the categories. However, as this finding is used to show that EDS is associated with comorbidity, they thus should discuss (the cause of) this finding in their discussion section in my view, if only to show their central finding is/seems real, or in the drawback session, indicating that this finding yet has to be explained.
---

REVIEWER	M.C. Scheper University of Applied Sciences Rotterdam, Institute for Health Innovation, Physical therapy, Rotterdam, the Netherlands The Hague university for Applied Sciences, faculty of health, nutrition and sports, medical health technology and innovation, the Hague, the Netherlands
REVIEW RETURNED	01-Aug-2019

GENERAL COMMENTS	Thank you for your responses. The revision has addressed my comments and I have no further additions.
---

VERSION 2 – AUTHOR RESPONSE

Reviewer: 1

Reviewer Name: Jacobs JWG

Institution and Country: UMCU, Netherlands

Please state any competing interests or state 'None declared': none

Please leave your comments for the authors below

The issues raised are in general addressed adequately. However, the authors should try to convince the readers of their paper more that their key findings of more additional codings and medication use

are real, not biased, as these findings are used to show that EDS is associated with comorbidity. This can be done in two ways.

1) I would suggest that the authors add the additional explanation given in their rebuttal on my previous issue 4 also to the paper, e.g. in the methods section.

4) the authors speak of "additional diagnoses", but these might be additional codings of the same diagnosis. For instance, in Figure 4, I wonder whether the high odds of congenital anomalies are additional codings of the congenital hypermobility syndrome; likewise, the category "musculoskeletal/connective tissue" with its high odds in cases seems additional codings of (consequences of) the hypermobility syndrome. This should be discussed as it is now an essential part of the conclusion.

Thank you for your comment. These are coded diagnoses excluding the diagnosis codes for JHS and EDS (see methods). Also, we counted people with these codes and not the numbers of codes themselves, therefore only one code in this category was selected for each person. Furthermore, the hypermobility syndromes do not come under the read-code sub-heading of congenital anomalies, so it is unlikely to be the explanation of this apparent difference. Some of these codes will be a direct reflection of comorbidities associated with EDS/JHS, but this is exactly the point we wanted to make and is part of our discussion.

The text now reads: "We then calculated odds ratios between cases and controls using Read chapters (excluding the Read codes for EDS and JHS)."

Thank you for your comment.

We have added the following text to the methods:

"We then calculated odds ratios between cases and controls using Read chapters (excluding the Read codes for EDS and JHS). This method counts the number of people with a code in each category; multiple codes for the same person in the same category are therefore not included. All results that affected at least 5 cases or 20 controls were visualised using forest plots. "

We have added the following text to the discussion:

"Although a snapshot of Read chapters codes that are more prevalent in our JHS/EDS cohort does not allow us to look at specific diagnoses and prescriptions, they can all be matched to conditions associated with EDS/JHS in the literature, for instance pain, fatigue, cardiovascular, gastrointestinal and gynaecological disorders, dysautonomia, mast cell activation as well as urinary tract infections ⁷. It needs to be stressed that these results exclude codes for EDS/JHS and that these are not part of the results for congenital anomalies or musculoskeletal conditions. We hope in future work to examine in greater detail these findings of significant differences between people with hEDS/HSD and others in order that we can better understand the nature of this condition, as well as potentially improving diagnostic recognition. "

2) My previous issue 5 was that the higher use of "Chemotherapy/Immunosupp drugs" is fully surprising to me and should be explained. The authors answer that they do not seek to explain, only to describe in a broad sense the differences between the categories. However, as this finding is used to show that EDS is associated with comorbidity, they thus should discuss (the cause of) this finding in their discussion section in my view, if only to show their central finding is/seems real, or in the drawback session, indicating that this finding yet has to be explained.

5) while it seems predictable that indeed cases used more "Musculoskeletal drugs" (Figure 5), and "Gastro-intestinal drugs" (co-medication of NSAID?), the higher use of "Chemotherapy/Immunosupp drugs" is fully surprising to me and should be explained. Moreover, what is the rationale of the lumping of the "Haematology/Dietetic" category?, what kind of drugs are these?

Thank you for your comment. We believe this might be connected to the use of azathioprine/ciclosporin/methotrexate for co-existent Crohns/colitis or rheumatoid disease, plus a few 'imab' drugs for auto-immune diseases such as lupus, or for rheumatoid arthritis, and some MS drugs (MS is thought to have a higher prevalence in this population) However, in this paper we do not seek to explain, only to describe in a broad sense the differences between the categories. We hope to undertake further work to dig down into the detail, and the reasons behind these findings, but we feel it would be premature to attempt to 'explain' our findings at this stage.

Haematology/dietetic drugs are both included in chapter 9 of the BNF, the model we used for our analysis. This is why they are shown together. They are mainly supplements, including iron, folic acid, vitamin B12, calcium and magnesium.]

The text now reads: "We hope in future work to examine in greater detail these findings of significant differences between people with hEDS/HSD and others in order that we can better understand the nature of this condition, as well as potentially improving diagnostic recognition. Having created this case-control cohort, further examination is made simpler as this first step has already been made."

Thank you for your comment.

We have added the following clarification to the text:

"We hope in future work to examine in greater detail these findings of significant differences between people with hEDS/HSD and others in order that we can better understand the nature of this condition, as well as potentially improving diagnostic recognition. For instance, immunosuppressant drugs are in the same Read chapter as chemotherapy drugs. Higher use of these prescriptions in the hEDS/HSD cohort could plausibly be linked to the known increased comorbid existence of disorders such as inflammatory bowel disease, inflammatory arthropathies, systemic lupus erythematosus and other autoimmune conditions, and is less likely to be due to a higher rate of use of chemotherapeutic agents. "